# Non-Coding RNAs in Multiple Myeloma Bone Disease Pathophysiology

**DOI:** 10.3390/ncrna6030037

**Published:** 2020-09-09

**Authors:** Lavinia Raimondi, Angela De Luca, Gianluca Giavaresi, Stefania Raimondo, Alessia Gallo, Elisa Taiana, Riccardo Alessandro, Marco Rossi, Antonino Neri, Giuseppe Viglietto, Nicola Amodio

**Affiliations:** 1IRCSS Istituto Ortopedico Rizzoli, SC Scienze e Tecnologie Chirurgiche–SS Piattaforma Scienze Omiche per Ortopedia Personalizzata, 40136 Bologna, Italy; angela.deluca@ior.it (A.D.L.); gianluca.giavaresi@ior.it (G.G.); 2Department of Biomedicine, Neurosciences and Advanced Diagnostics (Bi.N.D), Section of Biology and Genetics, University of Palermo, 90133 Palermo, Italy; stefania.raimondo@unipa.it (S.R.); riccardo.alessandro@unipa.it (R.A.); 3IRCCS ISMETT (Istituto Mediterraneo per i Trapianti e Terapie ad alta specializzazione), Research Department, 90127 Palermo, Italy; agallo@ismett.edu; 4Department of Oncology and Hemato-oncology, University of Milan, 20122 Milan, Italy; elisa.taiana@unimi.it (E.T.); antonino.neri@unimi.it (A.N.); 5Hematology, Fondazione Cà Granda IRCCS Policlinico, 20122 Milan, Italy; 6Institute for Biomedical Research and Innovation (IRIB), National Research Council (CNR), 90146 Palermo, Italy; 7Department of Experimental and Clinical Medicine, Magna Graecia University of Catanzaro, 88100 Catanzaro, Italy; rossim@unicz.it (M.R.); viglietto@unicz.it (G.V.)

**Keywords:** bone disease, long non-coding RNA, miRNA, multiple myeloma, non-coding RNA, tumor microenvironment

## Abstract

Bone remodeling is uncoupled in the multiple myeloma (MM) bone marrow niche, resulting in enhanced osteoclastogenesis responsible of MM-related bone disease (MMBD). Several studies have disclosed the mechanisms underlying increased osteoclast formation and activity triggered by the various cellular components of the MM bone marrow microenvironment, leading to the identification of novel targets for therapeutic intervention. In this regard, recent attention has been given to non-coding RNA (ncRNA) molecules, that finely tune gene expression programs involved in bone homeostasis both in physiological and pathological settings. In this review, we will analyze major signaling pathways involved in MMBD pathophysiology, and report emerging evidence of their regulation by different classes of ncRNAs.

## 1. Introduction

Multiple myeloma bone disease (MMBD) is a hallmark feature of multiple myeloma (MM), the second most common hematological malignancy characterized by abnormal proliferation of monoclonal plasma cells (PCs) in the bone marrow (BM). MMBD strikes approximately 80% of MM patients and causes debilitating bone pain, pathologic fractures, vertebral collapse and hypercalcemia, inducing significant patients’ morbidity and mortality [1].

The bone marrow microenvironment (BMM) is composed by a mineralized extracellular matrix and cellular components, including osteoclasts (OCs), osteoblasts (OBs), osteocytes (OCYs), immune cells, endothelial cells and stromal cells. Bone remodeling under pathological conditions is characterized by a strong inhibition of OBs activity, which leads to bone loss as OBs are unable to repair the lesions caused by the excessive osteoclastic resorption; the latter process is strongly supported by MM cells, which can exacerbate OCs activity promoting their maturation directly or by physically interacting with other cellular components, such as the BM stromal cells (BMSCs). In turn, cell–cell interactions and soluble factors or matrix-associated growth factors released from the resorbed bone increase MM cell proliferation and prompt tumor progression [2,3,4].

To effectively trigger BD, cellular components of the BMM produce and/or secrete a number of functional molecules, which collectively contribute to the osteoclastogenic events. In this regard, non-coding RNAs (ncRNAs) have recently emerged as fine regulators of gene expression programs underlying key molecular events featuring bone remodeling in MM.

Herein, we will briefly discuss the signaling pathways implicated in the development of MMBD, and, will then, analyze how they are modulated by manipulation or release of ncRNAs from different BMM cells.

## 2. Pathophysiology of MMBD

A great deal of literature has revealed that MMBD is regulated by a multiplicity of signaling pathways associated with anti-osteogenic, pro-osteoclastic and tumor-supporting properties [5]. Several intracellular and intercellular signaling cascades, as well as a large number of cytokines and chemokines, have been deeply studied and are nowadays considered valuable therapeutic targets in MMBD [6]. Dysregulation of signaling pathways by defective expression and/or function of ncRNAs has been implicated in MM pathogenesis, with an emerging role, also, in the onset of MMBD [7]. In this section, we will discuss about the main molecules and signaling pathways underlying MMBD pathophysiology; subsequently, available information about different ncRNAs known to affect relevant MMBD-related molecules and/or potentially involved in MMBD pathogenetic mechanisms will be provided.

### 2.1. RANK/RANKL/OPG Pathway

A key pathway regulating osteoclastogenesis is the RANK/RANKL/OPG signaling cascade. RANKL is expressed by BMSCs and OBs, while its receptor, the type I transmembrane protein RANK, is expressed by OCs precursor cells and mature OCs. RANKL/RANK interaction activates a complex signaling cascade, characterized by induction of the nuclear factor of activated T-cells, cytoplasmic 1 (NFATc1), which in turn regulates OC-specific genes, namely the tartrate-resistant acid phosphatase (TRAP), osteoclast-associated receptor (OSCAR) and cathepsin K (CTSK). Conversely, OPG is a soluble decoy receptor for RANKL. In physiologic conditions, a balanced RANKL/OPG ratio enables a correct bone remodeling, while in MM this ratio is strongly unbalanced, thus fostering bone destruction. MM PCs induce RANKL upregulation in OBs and BMSCs within the BMM; moreover, MM cells can express and secrete themselves RANKL [8]. Several studies evidenced RANKL upregulation in BM biopsies of MM patients and a positive correlation between the number of osteolytic lesions and increasing levels of serum RANKL [6,9,10].

### 2.2. Notch Pathway

Notch pathway includes four transmembrane receptors (Notch1-4) and five ligands (Jagged 1,2 and Delta-like 1,3,4). Following the receptor-ligand interaction, two proteolytic cleavages are mediated by ADAM/TACE and γ-secretase complex, which release the intracellular portion of Notch (ICN) to the nucleus where activates its target genes *Hes* and *Hey.* Notch pathway components are aberrantly expressed in MM cells and implicated in osteoclastogenesis and osteoblastogenesis processes occurring in the BMM; in particular, the roles played by Notch1, -2 and -3, as well as by Jagged1 and -2, have been described, indicating that not all signaling components are simultaneously involved in the same process or may even have opposing biological effects [11,12,13] Within the tumor microenvironment, the Notch/Ligand interactions can be homotypical or heterotypical. Notch signaling interferes with the maturation of the early OB pool by inhibiting the Wnt/β-catenin pathway in pre-OBs [14]. BMSCs also express Notch receptors that may be triggered by Jagged ligands of MM cells and subsequently increase RANKL production [1,15,16].

Osteoclastogenesis can be differently regulated by Notch signaling according to the various ligands and the receptor isoforms involved. For instance, Notch1 and Notch3 have been described as suppressors of OC differentiation [17], while Notch2 is upregulated during RANKL-induced early OC differentiation and involved in the late stage of osteoclastogenesis [18,19,20].

### 2.3. Wingless and Integration-1 (Wnt) Pathway

Wingless and integration-1 (Wnt) signaling is a master regulator of bone homeostasis, as it closely regulates the fine balance between bone-forming OBs and bone-resorbing OCs [21]. In the absence of Wnt, cytoplasmic β-catenin is bound and phosphorylated by a cytosolic complex constituted by the scaffold proteins APC, Axin1, the kinases GSK3 and CK1. Phosphorylation of β-catenin marks it for ubiquitination and proteasomal degradation [22]. The binding of Wnt with their cognate ligands, the Fz and LRP5/6 coreceptors, activates the signaling downstream; the receptor complex recruits the effector protein disheveled (Dvl), which in turn recruits Axin1-GSK3, thus blocking the cytosolic destruction complex. Hence, stable β-catenin translocates to the nucleus where, together with specific transcription cofactors, can activate the expression of Wnt target genes [23,24]. Regarding bone metabolism, Wnt signaling directs MSC differentiation towards differentiation into OBs [25]; furthermore, Wnt signaling promotes OBs survival, partly through the Src/ERK and PI3K/Akt pathways [21,26].

MM cells inhibit Wnt signaling and promote an OBs/OCs unbalance. MM cells and osteocytes express Wnt antagonists such as sclerostin, Dickkopf-1 (Dkk-1) and soluble frizzled-related proteins (sFRP-2/3), whose activity leads to OB suppression [27,28].

### 2.4. Dickkopf-1 (Dkk-1)

Wnt pathway can be also antagonized by Dkk-1, which plays an important role in osteoblastogenesis and skeletal development [27,29,30]. The binding of Dkk-1 to LRP5/6 receptors, in combination with the Kremen1/2 transmembrane proteins, induces the internalization of LRP and interferes with the activation of the canonical Wnt/β-catenin pathway [31]. Consequently, osteoblastogenesis and formation of the mineralized matrix are inhibited and, in turn, the undifferentiated BMSCs secrete IL-6 sustaining the proliferation of MM cells secreting Dkk-1 [32]. Dkk-1 also promotes osteoclastogenesis and bone resorption by modulating RANKL and OPG expression in OBs [33].

### 2.5. Sclerostin

Sclerostin (SOST), a cysteine knot-containing protein mainly produced by osteocytes, induces OB apoptosis by the caspase pathway and antagonizes the Wnt pathway by binding to the extracellular domain of LRP5/6 transmembrane receptors on osteoblast-lineage cells; moreover, sclerostin may prevent type I and type II bone morphogenetic proteins (BMPs) from binding to their receptors, thus reducing the BMP-mediated mineralization in OBs [34,35]. Sclerostin stimulates osteoclastogenesis also by increasing RANKL/OPG ratio [36], and its serum levels correlate with advanced MMBD and poor patient survival [37].

### 2.6. Bruton’s Tyrosine Kinase (BTK)

Another important pathway promoting osteoclastogenesis is downstream the Bruton’s tyrosine kinase (BTK), a non-receptor tyrosine kinase member of the Tec family also upregulated in MM PCs [38,39]. BTK inhibition reduced both tumor burden and osteolytic BD by decreasing OC number and activity, the adhesion of MM cells to BMSCs and the levels of BMSC-secreted growth factors [40]. Runt-related transcription factor 2 (Runx2), is critical in osteoblastogenesis and bone formation [41,42], and its upregulation correlated with an aggressive phenotype and poor prognosis of MM [43]. In MMBD, human MM cells inhibit OB formation and differentiation blocking the expression of Runx2/CBFA1.

### 2.7. Cytokines

MM is typically characterized by a desynchronized cytokine system with increased levels of pro-inflammatory cytokines [44]. A gene expression inflammatory signature could predict MM progression and patient survival and recently it has been identified an 8-genes signature (IL8, IL10, IL17A, CCL3, CCL5, VEGFA, EBI3 and NOS2) which accurately differentiates monoclonal gammopathy of undetermined significance (MGUS), smoldering myeloma (sMM) and MM [45]. MM cells can induce OCs differentiation and osteolytic activity, by modulating release from OBs of the proinflammatory cytokine IL-6 inside the tumor microenvironment; in turn, IL-6 inhibits OBs activity and induces the production of RANKL. Importantly, MM cells can induce BM adipocytes to make a more supportive niche and to increase OC activity through IL-6 and other molecules [46]. High levels of IL-8 are secreted from BMSCs of MM patients [47], which stimulated OC formation in vitro [48]. Macrophage inflammatory protein-1alpha (MIP-1α) is a cytokine with bone-resorbing properties secreted by MM cells and also by other BMM cells [49]; high expression of MIP-1α was found in BM PCs and in the serum of MM patients, and positively correlated with the presence of extensive lytic lesions and increased angiogenesis [50]. In addition, the inflammatory and bone-resorbing cytokine tumor necrosis factor (TNF)-alpha (TNF-α) is elevated in MM patients and correlated with MMBD. TNF-α sustains OC differentiation by directly targeting macrophages in a stromal environment expressing high levels of RANKL [51]. Furthermore, OCs produce B-cell activating factor (BAFF) and the proliferation-inducing ligand (APRIL), two members of tumor necrosis factor (TNF) family, which act as growth factors in MM cells, while MM cells in turn produce cytokines which stimulate the osteolytic activity in OCs [52].

In MM patients, upregulation of the cytokine activin A correlates with bone lesions and advanced disease. MM cells induce the production of activin A in BMSCs, partly through the JNK pathway. In turn, activin A inhibits OB differentiation by stimulating SMAD2 activity and inhibiting distal-less homeobox (DLX)–5 expression [53]. Blocking activin A signaling rescued MM-induced OB impairment, while reducing MM burden in a humanized murine model of MM [53]. Interestingly, activin A seems to be involved in bone remodeling also as an inducer of osteoclastogenesis, via stimulation of RANK expression and consequent enhancement of RANKL signaling [54].

An additional cytokine important in MMBD is the transforming growth factor β (TGFβ), that is produced in an inactive form by OBs in bone matrix and activated by OCs during bone resorption. During osteogenesis, TGFβ stimulates early OB proliferation, while blocking late-stage OB differentiation and mineralization to decrease bone formation [55]. TGFβ also increases bone lytic activity through stimulation of RANKL secretion and enhancement of OCs survival [56,57,58,59,60].

## 3. NcRNAs and MMBD

The non-coding compartment of the human genome represents almost the 98.5% of the whole human transcriptome. It has been widely demonstrated that ncRNAs critically regulate almost all physiologic and pathologic processes [61,62]. Based on their length, they have been classified into short (<200 nucleotides) non-coding RNAs (sncRNAs) or long (>200 nucleotides) non-coding RNAs (lncRNAs).

### 3.1. Short Non-Coding RNAs

MicroRNAs (miRNAs) are sncRNA molecules, of 17 to 24 nucleotides (nt) in length, that post-transcriptionally regulate mRNAs by perfect or partial complementarity to their 3′ untranslated region (3′ UTR), inducing either translational repression or degradation of target mRNAs. Since one miRNA can target hundreds mRNAs, it is obvious that these molecules have the capability to concomitantly regulate multiple pathways [63,64]. Dysregulation of miRNA expression and function has been shown to underlie the onset and progression of all cancer types, including PC dyscrasias [7]. Modulation of miRNA levels has been observed during MSCs differentiation both in physiological and pathological settings, with an emerging role also in the pathogenesis of MMBD. miRNAs involved in the MMBD pathophysiology and targeting the most relevant MMBD-related pathways will be discussed below. Figure 1 provides a graphic overview of mRNAs targeted by miRNAs and affecting osteoblastogenesis or osteoclastogenesis in MM.

#### 3.1.1. miR-221

The miR-221/222 cluster plays an oncogenic role in MM, where its inhibition induces significant anti-tumor activity by targeting key molecules involved in cell proliferation, survival and drug resistance as p27, p57 and PUMA [65,66,67]. The involvement of miR-221 in bone pathophysiology was initially suggested by its lower expression in osteoporotic compared with non-osteoporotic samples. Overexpression of miR-221 decreases the osteogenic potential of human mesenchymal stem cells (hMSCs), as indicated by the reduced expression levels of key OB markers, including osteocalcin (OC), alkaline phosphatase (ALP) and collagen, type I, α 1 (COL1A1); conversely, miR-221 inhibition led to the opposite effects. Biochemical experiments demonstrated that miR-221 targets Runx2, whose ectopic expression rescued miR-221 effect on OB markers, supporting the notion that miR-221-mediated OB differentiation occurs in a Runx2-dependent manner. miR-221-5p expression declined during OB induction of normal MSCs, while it remained unchanged upon differentiation of myeloma-derived MSCs. Notably, miR-221-5p inhibition increased the osteogenic differentiation capacity of MMBD-MSCs, and this effect was ascribed to SMAD3 down-regulation and to the activation of the PI3K/AKT/mTOR signaling pathway [68].

#### 3.1.2. miR-138

miR-138 was first identified as a negative regulator of hMSC OB differentiation. In vitro, miR-138 inhibition enhanced OB differentiation of hMSCs, whereas miR-138 overexpression inhibited their osteogenic potential. Moreover, miR-138 antagonism increased, whereas miR-138 overexpression reduced bone formation [69]. Increased expression of miR-138 was observed in MM cells and in MM MSCs compared to that from healthy subjects; co-culture with MM cells upregulated miR-138 in healthy MSCs, suggesting that the interplay between MM cells and MSCs drives the dysregulated miR-138 expression in MSCs.

Interestingly, inhibition of miR-138 with an LNA-modified anti-miR-138 oligonucleotide was able to enhance the osteogenic differentiation of MSCs in vitro; moreover, bone formation rate and OBs number were significantly increased in MM bearing mice treated with anti-miR-138 LNA, indicating that miR-138 negatively regulates bone apposition in MM. Gene set enrichment analysis performed on anti-miR-138-treated cells revealed that the regulation of chondrocyte differentiation gene set was enriched in the OBs inhibited for miR-138, suggesting that miR-138 antagonism induces bone formation in the context of MM by de-repressing target genes involved in osteochondrogenesis. Three putative miR-138 targets known to be important for the induction of osteogenic and chondrogenic MSC differentiation, namely ROCK2, TRPS1 and SULF2, were de-repressed after anti-miR-138 treatment [70].

#### 3.1.3. miR-203a-3p.1

During osteoblastogenesis, miR-203a-3p.1 levels were found to decline in normal MSCs, whereas no change was observed in MM MSCs. In line with these findings, the authors demonstrated that canonic OB differentiation markers, including ALP, OPN and OC, were upregulated in MM-MSCs following treatment with anti-miR-203a-3p.1 oligonucleotides. The inhibitory effects on hMSCs osteogenic activity by miR-203a-3p.1 was likely dependent on the targeting of SMAD9 and of Wnt/β-catenin pathway, which promote OB differentiation and bone formation; accordingly, rescue experiments confirmed the key role of SMAD9 down-regulation in miR-203a-3p-mediated osteoblastogenesis [71].

#### 3.1.4. miR-21

miR-21 is an established onco-miRNA in MM [72], where its expression is induced by IL-6 in a STAT3-dependent manner [73]. miR-21 was found upregulated in OCs [74] and in BM mononuclear cells of MM patients [75], supporting a role within the BM milieu. We previously showed that miR-21 plays a pivotal role in sustaining MMBD by regulating RANKL/OPG ratio in the MM BM microenvironment. Higher levels of miR-21 were found in MM BMSCs as compared with healthy BMSCs. Importantly, we validated OPG as a direct target of miR-21 and reported that selective inhibition of miR-21 in MM BMSCs was able to restore OPG expression and secretion and to reduce RANKL levels. As a result, miR-21 inhibition in BMSCs suppressed the bone lytic activity of OCs in vitro, thus pointing to miR-21 as candidate target to treat MMBD [76]. In MM, Th17 cells sustain tumor growth and OCs-dependent bone damage. We found that Th17 from MM patients with BD express significantly higher miR-21 levels as compared to non-osteolytic MM and healthy controls; importantly, early inhibition of miR-21 in naive T cells impaired Th17 differentiation in vitro and abrogated Th17-mediated MM cell proliferation and OCs activity. These findings were recapitulated in vivo in NOD/SCID-γ-NULL mice intratibially injected with T cells transfected with miR-21 synthetic inhibitors and MM cells. At a molecular level, a pairwise RNAseq and proteome/phosphoproteome analysis demonstrated that miR-21 inhibition in Th17 cells upregulated STAT-1/-5a-5b, impaired STAT-3 and redirected Th17 towards Th1/Th2 like activated/polarized cells [77].

#### 3.1.5. miR-29b

Several reports indicate that the expression of miR-29 family members is widely deregulated in hematologic malignancies [78,79], and their reconstitution deeply impacts on the phenotype of cancer cells through the targeting of epigenetic regulators [80,81,82]. miR-29b was found upregulated along osteoblastogenesis during late mineralization phases. Specifically, miR-29b overexpression turned off the expression of key OB-inhibitory proteins such as TGFβ3 and HDAC4, resulting in Runx2 upregulation, and also down-regulated the Wnt pathway inhibitor catenin beta interacting protein 1 (CTNNBIP1) [83]. In parallel, our studies defined the role of miR-29b in the inhibition of OCs generation and function. In fact, we found that miR-29b was progressively down-regulated during OCs differentiation of monocyte precursors under M-CSF and RANK-L stimulation, and restoration of miR-29b expression in OCs precursors strongly antagonized OCs resorbing activity by reducing intracellular levels of TRAP, cathepsin K, metalloproteinase type 2 and 9 (MMP-2 and MMP-9). Reduced expression levels of such OC-resorbing enzymes were due to the targeting of MMP-2 and c-FOS mRNAs, which in turn impaired the rearrangement of actin rings, whose normal morphology is critical for OCs resorbing activity and bone adherence. Enforced expression of miR-29b also reduced RANK expression on the cell surface, hampering OCs response to RANKL stimulation and reducing in vitro osteoclastogenesis induced by MM cell lines [78,84]. Importantly, miR-29b levels were significantly reduced in MM dendritic cells, and its reconstitution counteracted pro-inflammatory pathways in co-cultured MM cells, including signal transducer and activator of transcription 3 and nuclear factor-κB, and cytokine/chemokine signaling networks, which correlated with patients’ adverse prognosis and development of BD. Moreover, miR-29b downregulated IL-23 in vitro and in the SCID-synth-hu in vivo model, and antagonized a Th17-driven inflammatory response, which notably sustains MM cell growth and osteoclast-dependent bone damage [85].

#### 3.1.6. miR-214

miR-214 was found implicated in physiological processes, as osteogenesis, osteoclastogenesis and muscle development, as well as in pathological conditions. Misiewicz–Krzeminska et al. demonstrated the down-regulation and the tumor suppressor roles of miR-214 in MM cells. Restoration of miR-214 expression level in MM cells enhanced apoptosis and the inhibition of cellular proliferation, through the inhibition of p53/MDM2 interaction and of the DNA replication pathway [86]. By analyzing the expression profile of miRNAs on a large cohort of MM patients with different stages of BD, Hao and colleagues showed that circulating miR-214 in the serum of MM patients significantly correlated with the degree of bone injury, and MM patients with a higher level of serum miR-214 had a poor outcome. In addition, miR-214 levels in MM patients with lytic bone lesions were higher than those without bone lesions. Indeed, patients with high serum miR-214 levels showed a significant reduction in progression-free survival (PFS) and overall survival (OS). On the other hand, when treated with bisphosphonates MM patients presenting higher miR-214 serum level benefited from a significant increase in their quality of survival, with effects on reduction of bone lesions rather than on tumor burden. According to these data, miR-214 levels in patients serum may be used as biomarkers for the detection of MMBD, as well as prognostic markers for MM patients with BD to define the start of treatment with bisphosphonates [87].

#### 3.1.7. miR-135b

Abnormal expression of miR-135b is reported in different types of tumors, with a clear role in tumorigenesis. As reported by Xu et al., miR-135b is involved in MMBD impairing the osteogenic differentiation capability of BM-derived MSCs from MM patients (MM-hMSCs) by targeting at 3′ UTR SMAD5, which is involved in osteogenesis. Upregulation of miR-135b in MM BM-derived hMSCs causes a reduction of ALP activity. On the other side, miR-135b inhibition in MM-hMSCs maintained in osteogenic medium restored the activity of ALP and of other osteogenic markers (BSP, COLA1 and OPN). The increase of miR-135b also was observed during co-culture of hMSCs from healthy donors with MM cells lines, indicating that the MM microenvironment modulates the miRNome of hMSCs reducing their osteogenic potential [88]. This may explain why Hao and colleagues observed dysregulated levels of miR-135b in the serum of the MM patients, which could be used to distinguish patients with or without bone lytic lesions [87]. Therefore, miR-135b inhibitors may represent potential RNA therapeutics to direct hMSC osteogenic differentiation towards bone formation [88].

#### 3.1.8. miR-342 and miR-363

High levels of the bone-specific transcription factor Runx2 is observed in several solid tumors, promoting bone metastasis and osteolysis [43,89]. Recently, some studies have reported that MM cells release soluble factors to suppress osteoblastogenesis through Runx2 inhibition in immature and pre- OBs at new bone sites [90]. By comparing the plasma of MM patients with normal donors, Gowda et al. showed that high levels of Runx2 in MM cells inversely correlated with miR-342 and miR-363 expression. Overexpression of miR-342 and miR-363 in CAG MM cells, individually or in combination, suppressed Runx2 protein levels, with concomitant inhibition of Akt/β-catenin/survivin, known targets of Runx2, leading to a strong reduction of MM cell proliferation. The combined miRNAs action on MM cells also led to a decrease of DKK1 and RANKL levels both in in vitro and in vivo preclinical models. In MM-bearing mice, miR-342 and miR-363 acted on the BMM by increasing OB and decreasing OC numbers, antagonizing bone resorption in vivo. Moreover, overexpression of miRNAs led to an improvement of anti-tumor immunity in vivo, increasing the number of immunosuppressive regulatory T and B cells and decreasing dendritic cells within the tumor microenvironment [89].

#### 3.1.9. miR-223

miR-223 was identified as a key regulator for differentiation of myeloid precursors and OCs. In cancer, deregulation of miR-223 was described in leukemia and lymphomas [91,92]. Evidence of a miR-223 tumor supportive role was reported in MM, where its expression in MSCs decreased in a cell-adhesion dependent manner [93]. Co-culture of MM cells and MM-MSCs induced activation of Notch signaling via jagged-2/notch-2, leading to increased expression of *Hes1*, *Hey2*, or *Hes5* in both cell types; in turn, activation of Notch signaling in MM-MSCs led to a decrease in miR-223 expression, although the underlying mechanism was not addressed. To support role of Notch signaling on the miR-223 decrease, the authors demonstrated that Notch pathway inhibition either through the γ-secretase inhibitor GSI-XII or a jagged-2 neutralizing antibody, upregulated miR-223 levels. Of note, this study also highlighted the role of miR-223 in the osteogenic differentiation of MSCs; in fact, miR-223 levels significantly increased when osteogenic differentiation was induced. Moreover, inhibition of miR-223 expression impaired the osteogenic differentiation potential of MSCs, decreasing the expression of Runx2 and osteopontin, and reducing ALP activity and calcification [93].

### 3.2. Long Non-Coding RNAs

Long non-coding RNAs refer to a highly heterogeneous class of non-coding RNAs longer than 200 nt, that include intergenic transcripts (lincRNAs), enhancer RNAs (eRNAs) and sense or antisense transcripts that overlap other genes [94]. They have been found implicated in several molecular functions, including in cis or in trans transcriptional regulation, organization of nuclear domains and regulation of proteins and/or RNA molecules; functionally, they can act as guides for ribonucleoprotein complexes, dynamic scaffolds and molecular decoys for proteins including transcription factors and miRNA sponges [7,94,95,96].

Overall, lncRNAs play crucial roles in several cellular processes, such as DNA repair, proliferation, angiogenesis and epithelial-mesenchymal transition; moreover, their expression is linked with various diseases and some of them have been described as potential disease biomarkers [97,98,99,100,101]. Indeed, numerous lncRNAs have been described as aberrantly expressed in various cancers and/or associated with clinically relevant cancer subtypes, predicting tumor behavior and prognosis [102,103]. Recent studies have clarified the expression profiles of lncRNAs in PC dyscrasias [7,104,105]. In MM, three distinct transcriptomic analyses unraveled a dysregulated lncRNA landscape. Ronchetti et al. used microarray technology to analyze lncRNA expression in patients at different stages of MM progression—including MGUS, sMM, MM and PCL—and in healthy donors, identifying 31 lncRNAs altered in tumor samples [106]. RNA-seq was used by the same authors in a follow-up study, to evaluate lncRNA expression in 30 MM patients, leading to the identification of 391 dysregulated lncRNAs, and also extended the study to the main MM molecular subgroups and genetic alterations [107]. Samur et al. instead described the lncRNA landscape in MM cells by RNA-seq on PCs from 308 newly-diagnosed and uniformly treated MM patients enrolled to the DFCI/IFM 2009 clinical study, and developed a prognostic model based on lncRNAs to stratify patient risk [108].

Since several lncRNAs have been already described for their role in bone metabolism and osteogenesis, they have also been investigated for their potential role in MMBD [104]. Collectively, functional investigations on lncRNAs in the context of MMBD are limited respect to miRNAs, and this gap is likely dependent on their complex 3D structure interacting with various molecular partners, leading to a pleiotropic activity which makes them difficult to study [7].

A recent study evaluated 17 lncRNAs, whose targets were described as crucial molecules in MM and bone homeostasis; analysis was carried out in MM patients with bisphosphonate-induced osteonecrosis of the jaw (BONJ), in MM patients without BONJ and in a group of healthy controls. Results obtained evidenced a different lncRNA profile in BONJ patients compared to MM patients and controls. Notably, two lncRNAs (DANCR and metastasis associated in lung adenocarcinoma transcript 1 (MALAT1)) resulted downregulated when compared to controls and MM, while twelve were found overexpressed in MM with BONJ. Overall, the authors suggested that targeting these lncRNAs could represent a valuable tool for prevention and therapy of BONJ [109].

The most relevant lncRNAs found which are potentially implicated in MMBD pathophysiology are below summarized.

#### 3.2.1. LncHOXC-AS3

As stated above, MMBD is characterized by severely impaired osteogenesis [5]. The lncRNA HOXCAS3 is transcribed in opposite to HOXC10 and positioned at chromosome 12q13.13; in MM it is expressed in MM MSCs.

HOXC-AS3 interacts with HOXC10 at the overlapping parts thus strengthening its stability and promoting its expression; consequently, up-regulation of HOXC10 contributed to repressing osteogenic potential of MSCs [110]. The effect of HOXC-AS3 on osteogenic differentiation was evaluated in vivo, by using a “human-in-mouse” xeno-transplantation MM model, which included the MM clinical signs such as BD. To analyze bone loss in the mouse models, trabecular and cortical bone were evaluated by micro CT images; results obtained indicated severe bone loss in the vehicle control group, compared to the HOXC-AS3 siRNA-treated mice group. In addition, HOXC-AS3 siRNA significantly altered bone turnover markers levels in serum; in particular, the bone resorption marker C-terminal telopeptide of type 1 collagen (CTX) was reduced in HOXC-AS3 siRNA group, while the bone formation marker, procollagen type 1 N-terminal propeptide (P1NP), increased [110].

#### 3.2.2. TUG1

The taurine upregulated gene 1 (TUG1) lncRNA contributes to the formation of photoreceptors and plays crucial roles in retinal development [111]. TUG1 is a 7.1 kb lncRNA transcribed from human chromosome 22q12.2. Different works have reported that TUG1 has pro-tumor activity in several tumors, such as osteosarcoma, melanoma, cholangiocarcinoma [112], glioma and hepatocellular carcinoma, non-small cell lung cancer and bladder cancer [113]. Recently, through qRT-PCR analysis of serum samples from 98 healthy controls and 110 MM patients, Qingqing et al. observed increased TUG1 levels in MM. To explore the diagnostic utility of TUG1, they compared TUG1 expression levels in different stages of disease in MM patients, alone or together with other diagnostic biomarkers such as albumin and β2-microglobulin. Univariate analyses confirmed the correlation of TUG1 levels with different disease stages in MM, observing a higher serum TUG1 level in patients with bone lesion. Furthermore, the multivariate analysis combining all three biomarkers significantly improved the AUC value of ROC analysis, which was better than any individual marker analysis. These results suggested the usefulness of TUG1 as a potential diagnostic biomarker suitable for clinical use [114], although its functional role within MMBD remains to be investigated.

#### 3.2.3. MALAT1

The lncRNA metastasis associated in lung adenocarcinoma transcript 1, which is located on the human chromosome 11q13, has a total length of about 8 kb. MALAT1 regulates the transcriptional and translational levels of proto-oncogene Runx2 in colorectal cancer metastasis [115]. It is expressed in different tissues where it regulates gene expression, alternative splicing and cell cycle. MALAT1 has been found overexpressed in several human neoplasms and promotes tumor cell invasion and metastasis [116,117]. Cho et al. first investigated the relationship between MALAT1 and MM disease. They evaluated MALAT1 expression by comparing the mononuclear bone marrow cells of MM patients in different disease states compared to healthy individuals, and demonstrated that the high serum levels of MALAT1 detected in MM patients decreased significantly in post-treatment patients, showing serum levels similar to those of healthy individuals and a prolonged progression-free survival [118]. We studied MALAT1 functional role in MM and demonstrated that it may promote cell survival by regulating the expression and activity of the proteasome machinery [119]. Regarding its role in normal bone homeostasis, Xiao et al. demonstrated that MALAT1 promotes osteoblastogenesis via miR-204 sponging, and upregulating the miR-204-target SMAD4, which in turn promoted the expression of ALP and osteocalcin responsible of the increased bone formation and mineralization [120]; however, MALAT1 functional role in MMBD has not been, so far, investigated.

#### 3.2.4. MEG3

Zhuang et al. found that the lncRNA maternally expressed gene 3 (MEG3), which is located on 14q32.2, promotes the differentiation of MM BMSCs into OBs. Mechanistically, MEG3 promotes the translation of the downstream BMP4 gene by preventing the inhibitory effect of *SOX2* on the BMP4 promoter [121]. Conversely, Li et al. showed that MEG3 is downregulated during adipose-derived MSC differentiation into adipogenic cells, while upregulated during osteogenic differentiation. Interestingly, MEG3 silencing promoted the osteogenic and adipogenic differentiation of human adipose-derived MSCs [122]. Additional studies are indeed required to clarify the role of MEG3 in MMBD pathophysiology.

### 3.3. Circular RNAs

Circular RNAs (circRNAs) are covalently closed single-stranded lncRNAs whose expression is finely regulated in a tissue- and disease-specific manner. They can act as modulators of transcription, as miRNA sponges or can even sequester factors, such as RNA-binding proteins or ribonucleoprotein complexes [123]; of note, they can be also packaged and released into extracellular vesicles (EVs) [124]. Emerging evidence has highlighted that circRNAs are implicated in various malignancies and can function as potential diagnostic and prognostic biomarkers [125,126,127,128]. High-throughput RNA sequencing has recently revealed thousands of circRNAs expressed in different cancer cell lines [129]. Collectively, few studies have been addressed in MM [130,131,132,133]. Dahl et al. [127] identified 619 unique circRNAs in the MM cell line NCI-H929. By RNA-seq, Liu et al. identified circRNAs differentially expressed in MM patients respect to the healthy individuals, focusing on hsa_circRNA_101237, that positively correlated with some clinical features of MM patients, including bone destruction [132]. Upregulation of hsa_circRNA_101237 in recurrent/refractory patients compared to first-episode treatment-naive patients was shown; similar results were obtained after analysis of expression levels of hsa_circRNA_101237 in different MM cells lines, with the highest expression levels of hsa_circRNA_101237 observed in bortezomib resistant cell lines. They also analyzed the correlation between hsa_circRNA_101,237 expression levels and MM types or bone lesions, chromosomal variations and genetic variations. Interestingly, it was found that hsa_circRNA_101237 was overexpressed in positive patients for 13q14 deletion, 1q21 amplification, P53 deletion and t(4,14) and t(14,16), along with KRAS, NRAS, FAM46C, DIS3, BRAF, TRAF3 and TP53 gene mutations, associated with poor prognosis of MM patients. In particular, the overexpression of hsa_circRNA_101237 was closely related in MM patients with cytogenetic abnormalities and BD, which were of R-ISS stage III, defining it as a marker of high clinical value [132]. The mechanistic role of this circRNA in the context of MM and MMBD is yet to be analyzed.

## 4. Extracellular Vesicle-Associated ncRNAs

It has been widely demonstrated that different ncRNAs species are contained in extracellular vesicles, which are lipoproteic structures heterogeneous in size and content, released by almost all cell types [134,135]. In recent years, EVs have gained attention because of the identification of biological molecules as cargo. In fact, if they were initially considered a way of elimination of waste products [136], current knowledge indicates that EVs represent a cell–cell means of communication.

EVs play a crucial role in the context of MM pathobiology, and specifically in the crosstalk that malignant PCs establish with other cells of the BMM such as endothelial, stromal, MSCs and immune cells [137,138,139,140,141,142]. Such interaction is key both in the progression of the disease and in the onset of pharmacological resistance [143,144]. Moreover, growing experimental evidence indicates that EVs released by MM cells alter bone homeostasis and, therefore, contribute to the onset of MMBD [145,146,147,148,149].

We firstly demonstrated that EVs were released from MM cell lines and were also detectable in the serum of MM patients. Such EVs induced the osteoclastic differentiation of murine macrophages as well as of human pre-osteoclasts, enhancing the expression of specific OC differentiation markers, such as CTSK, MMP9 and TRAP. MM-EVs were able to induce a complete osteoclastic differentiation. Notably, pre-osteoclast treated with MM-EVs differentiated in multinuclear and giant OCs, having a strong erosive capability, as evidenced by bone resorption pit assays; these effects were not observed when EVs derived by the metastatic colorectal cancer cell line SW620 were used, demonstrating the MM cell-type specificity of EVs within the BMM [145].

Subsequently, many studies have disclosed the molecular mechanisms underlying the EV-dependent osteoclastogenic effect. In 2019, Raimondo et al. showed that the presence of the EGFR ligand, amphiregulin (AREG), partially mediates the EV-mediated OC activation. Authors observed that AREG was specifically enriched in exosome samples, leading to the activation of EGFR in pre-OCs; such effects were abrogated by exosome pre-treatment with anti-AREG neutralizing Ab [146].

In parallel, the role of EVs on another crucial cell population involved in bone homeostasis, i.e., OBs, has been investigated. The results of these studies indicate that MM EVs can inhibit the osteogenic differentiation of MSCs, thus contributing to increased osteolysis [146,147,148,150,151,152]. In one of these studies, it emerged that MM EVs carry DKK1 in OBs leading to reduced levels of Runx2, osterix and collagen 1A1 [152].

Noteworthy, increasing evidence suggests that the EV-associated ncRNA cargo mediates the profound impact that EVs exert on the gene expression profile of target cells [153,154,155].

Research has been performed to identify the mechanisms underlying the specific sorting of certain ncRNAs in EVs. For some miRNAs, a small sequence, called hEXO motif, has been identified as recognized by the RNA binding protein SYNCRIP, and found to be responsible for the specific sorting of these miRNAs in vesicles [156].

Recent evidence also correlated the osteolytic effect of MM-EVs with its ncRNAs content. A study published by Li et al., showed that the lncRNA RUNX2-AS1 is packaged in MM-EVs and transferred to MSCs, resulting in the transcriptional repression of Runx2 and, thus, prematurely blocking MSCs osteogenic differentiation [147]. Finally, by analyzing the miRNA repertoire of MM-EVs, several miRNAs involved in the inhibition of osteogenic differentiation [148,149] were identified. In a recent study, miR-103a-3p was identified as one of the upregulated miRNAs following the treatment of MSCs with MM-EVs [148]; in parallel, miR-129-5p, carried by MM-EVs, was found to reduce ALPL levels in MSCs. In addition, the authors also observed that miR-129-5p was more abundant in EVs isolated from MM patients with active BD than in SMM, supporting the notion that this miRNA is key in mediating the EV-dependent MMBD [149].

A more in-depth characterization of the ncRNAs contained in these structures will allow the development of new biomarkers for the diagnosis and prognosis of the disease. In parallel, the identification of the mechanisms underlying the exact sorting of ncRNAs, and/or other biomolecules, in EVs, could provide the basis for the definition of new therapeutic targets against MMBD.

## 5. Conclusions

Preclinical studies have clarified the pathophysiology of MMBD identifying new possible druggable targets, which could hopefully enlarge the spectrum of therapeutic opportunities against this severe MM complication. In this context, a novel area of investigation is represented by ncRNA molecules, which can finely tune the gene expression programs of BMM cells finally regulating bone homeostasis. ncRNAs have been found to be expressed and secreted within EVs by various cellular components of the BMM. By modulating the expression of signaling molecules implicated in OC development, ncRNAs have been demonstrated to control different processes in OCs precursors and mature OCs. A graphic overview of the molecular and cellular scenario underlying MMBD pathophysiology and overall reviewed in this manuscript is reported in Figure 2. Collectively, available literature findings presented in this review point to a promising role of these molecules as novel therapeutic and prognostic tools in MMBD, able to affect the expression of some of the most important MMBD-related pathways. However, it has to be noticed that although the prognostic relevance of ncRNAs different from miRNAs (i.e., lncRNAs, circRNAs) is progressively emerging, functional studies clarifying their biological activity are at their infancy or still missing and need to be mandatorily performed to dissect their precise role in MMBD. Moreover, lncRNA expression profiling of BMM cells under specific differentiation stimuli is warranted to identify useful lncRNA signatures to be functionally characterized in the context of MMBD.

The most relevant ncRNAs functionally implicated in MMBD, their main molecular targets and their bone-related effects are summarized in Table 1.

## Figures and Tables

**Figure 1 ncrna-06-00037-f001:**
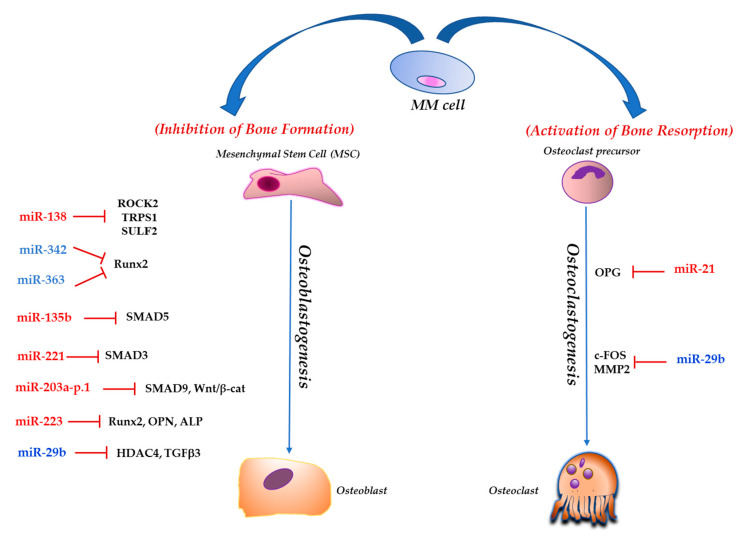
The picture reports microRNAs (miRNAs) and their corresponding mRNA targets involved in osteoblastogenesis and osteoclastogenesis processes in the multiple myeloma (MM) bone marrow microenvironment (BMM). miRNAs triggering bone resorption are reported in red; miRNAs triggering bone apposition are reported in blue.

**Figure 2 ncrna-06-00037-f002:**
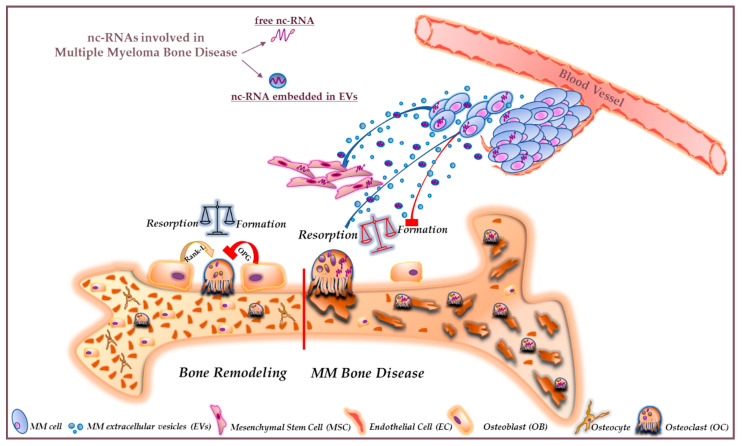
Cartoon showing the effect of dysregulated non-coding RNAs (ncRNAs) (miRNAs and long (>200 nucleotides) non-coding RNAs (lncRNAs)) on multiple myeloma bone disease (MMBD). ncRNAs modulate bone remodeling both in physiologic conditions (on the left side of the red line across the bone) or in MM (on the right side of the red line across the bone). Under physiologic conditions, osteoclasts (OCs) and osteoblasts (OBs) work together to balance bone resorption and formation. Their activities are governed by specific gene programs, whose expression is regulated by ncRNAs, i.e., miRNA or lncRNAs produced by MM cells and/or other cells of the BMM, and released within extracellular vesicles (EVs). In MMBD, dysregulated ncRNAs contribute to enhance OCs activity for instance acting on RANKL/OPG pathway, while OB activity is inhibited, thus establishing an unbalanced condition that fosters the development of osteolytic lesions.

**Table 1 ncrna-06-00037-t001:** Molecular targets and bone-related effects of functionally characterized ncRNAs in MMBD.

ncRNAs	Target(s) in MMBD	Bone-Related Effects	Function in MMBD	Reference(s)
hsa-miR-221	SMAD3	Decreases the osteogenic potential of hMSCs	oncomiRNA	[68]
hsa-miR-138	ROCK2, TRPS1 and SULF2	Decreases the osteogenic and chondrogenic potential of hMSCs	oncomiRNA	[69,70]
hsa-miR-203a-3p.1	SMAD9 and Wnt/β-catenin pathway	Decreases the osteogenic potential of hMSCs	oncomiRNA	[71]
hsa-miR-21	OPG	Regulates RANKL/OPG ratio in the MM BM microenvironment	oncomiRNA	[76,77]
hsa-miR-29b	c-FOS; MMP2	Negatively regulates human OCs differentiation and function	TS miRNA	[84,85]
hsa-miR-135b	SMAD5	Impairs the osteogenic differentiation capability of BM-derived MSCs from MM patients	oncomiRNA	[88]
hsa-miR-342 and miR-363	Runx2	Impact the BMM decreasing OBs activity and increasing OCs activity	TS miRNA	[89]
hsa-miR-223	Runx2 Osteopontin	Impairs the osteogenic differentiation potential of MM-BMMSCs	TS miRNA	[93]
HOXC-AS3	HOXC10	Represses the osteogenic potential of MSCs	Oncogenic lncRNA	[110]
MEG3	BMP4	Promotes the differentiation of MSCs into OBs	TS lncRNA	[121]

TS, tumor suppressor.

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
