# Peer review of "Non-Coding RNAs in Multiple Myeloma Bone Disease Pathophysiology"

_ncrna, 2020, doi:10.3390/ncrna6030037_

Round 1
Reviewer 1 Report
The Authors have chosen a relevant and important topic to review. Role of non-coding RNAs are now all pervasive and it is only likely that they play a significant role in Multiple Myeloma Bone Disease (MMBD). I would recommend reorganizing the material significantly for a better focus and readability.
The authors have themselves identified the focal point of the review as signalling pathways that influence MMBD pathophysiology. This is well thought of and a good strategy. Most ncRNAs will ultimately function by regulating protein-coding transcript and thereby regulating biological pathways. But currently the ncRNA part of the review is not well connected with the section of signalling pathways. Thus, no clear picture emerge. I suggest the following:
- Use different signalling pathways as sub-headings for different sections and bring in the mRNAs and the ncRNA regulating those mRNAs within that section. For example, the review describes Notch signalling pathway is an important pathway involved in MMBD. But it does not describe ncRNAs that regulate mRNAs involved in Notch signalling. That connection needs to be made.
- If there are important ncRNAs (e.g. lncRNAs) which are well established to have a role in MMBD but not known which pathways they influence then have a separate section for those ncRNAs which does not belong to any relevant signalling pathway
- If there are important microRNAs with established role in MMBD but their mRNA targets unknown or does not fall in signalling pathways, then perform a target prediction analysis and add that data as a predictive model
- Write about the expression profile of the ncRNAs with respect to the signalling pathways and MMBD pathophysiology. If it is NOT known whether the discussed ncRNAs are expressed in the relevant cell/tissue type, then include it as a limitation and future research scope.
- Create figures for pathways and bring in a multi-layered approach where mRNA and their regulator ncRNAs are depicted. The current figure is good but its not specific enough
- Add the word 'multiple' in the title before 'myeloma'
Author Response
# REVIEWER 1
The Authors have chosen a relevant and important topic to review. Role of non-coding RNAs are now all pervasive and it is only likely that they play a significant role in Multiple Myeloma Bone Disease (MMBD). I would recommend reorganizing the material significantly for a better focus and readability.
The authors have themselves identified the focal point of the review as signalling pathways that influence MMBD pathophysiology. This is well thought of and a good strategy. Most ncRNAs will ultimately function by regulating protein-coding transcript and thereby regulating biological pathways. But currently the ncRNA part of the review is not well connected with the section of signalling pathways. Thus, no clear picture emerge. I suggest the following:
1) Use different signalling pathways as sub-headings for different sections and bring in the mRNAs and the ncRNA regulating those mRNAs within that section. For example, the review describes Notch signalling pathway is an important pathway involved in MMBD. But it does not describe ncRNAs that regulate mRNAs involved in Notch signalling. That connection needs to be made.
Re: We thank the reviewer for this comment. We modified the paragraph ’Pathophysiology of MMBD’ reporting, as sub-headings, the different molecules and signaling pathways involved in MMBD.
Since information on ncRNAs targeting MMBD-related molecules is constantly growing and research in this field is a relatively new area of investigation, it emerges that not all the signaling pathways involved in MMBD have been so far investigated for their regulation by ncRNAs. Therefore, in our manuscript we have decided to provide first an overview of the major signaling pathways implicated in MMBD pathophysiology, and then to individually present all the ncRNAs found to impact on MMBD, discussing-if any- molecular and functional data emerging from the experimental investigations retrieved from the literature. We have clarified this point in page 2, lines 65-69.
We have also added a new Figure (Fig. 1), which depicts the molecular targets of the functionally investigated miRNAs acting in osteoblastogenesis and osteoclastogenesis processes in the myeloma bone marrow microenvironment, which are also summarized in Table 1 along with bone related effects and the tumor suppressive/promoting role in the context of MMBD.
Moreover, we have added a paragraph about miR-223 role in MMBD and its regulation by the Notch pathway.
2) If there are important ncRNAs (e.g. lncRNAs) which are well established to have a role in MMBD but not known which pathways they influence then have a separate section for those ncRNAs which does not belong to any relevant signalling pathway.
Re: Similar to what reported for miRNAs, we have individually analyzed the lncRNAs known to impact on MMBD. However, we have stressed in the paragraphs related to each lncRNA as well as in the conclusions section, that research about the functional role of lncRNAs and circRNAs is still at its infancy, and further studies are needed to shed light on their precise effect in MMBD.
3) If there are important microRNAs with established role in MMBD but their mRNA targets unknown or does not fall in signalling pathways, then perform a target prediction analysis and add that data as a predictive model.
Re: The greatest part of miRNAs with a functional role in MMBD, such as miR-29b, miR-21, miR-138 and others listed in Table 1, have been analyzed with regard to their molecular targets in the context of the MMBD, and also reported in the new Fig. 1. Regarding miR-214, although the tumor suppressive role in MM plasma cells, as well as it prognostic role in MMBD patients treated with bisphosphonates have been reported, molecular correlates of miR-214 effects on MMBD have not been investigated by the authors.
However, we feel beyond the scope of this review to identify unknown mRNA targets of miRNAs, like miR-214, potentially involved in MMBD. Moreover, given that mRNA targeting at 3’UTR by miRNAs is highly cell type specific, in silico findings would need to be mandatorily validated by additional assays (like 3’UTR-luciferase reporter assay) in order to confirm their effects on cells of the BMM, and more in general on MMBD. Regarding lncRNAs, given the multitude of mechanisms of action, biochemical studies to identify protein/RNA interactors and/or affected gene loci are necessary to finally dissect molecular targets and pathways affected by such molecules in the context of MMBD.
4) Write about the expression profile of the ncRNAs with respect to the signaling pathways and MMBD pathophysiology. If it is NOT known whether the discussed ncRNAs are expressed in the relevant cell/tissue type, then include it as a limitation and future research scope.
Re: Thank you for this observation. Most of the reported studies indicate miRNA expression deregulation along osteoblastic or osteoclastic differentiation. Information for each specific miRNA has been reported in the corresponding sub-paragraph. We have also added, in the conclusions section, that lncRNA expression profiling of BMM cells under specific differentiation stimuli could provide useful information on specific lncRNA signatures potentially involved in MMBD to be functionally characterized.
5) Create figures for pathways and bring in a multi-layered approach where mRNA and their regulator ncRNAs are depicted. The current figure is good but its not specific enough
Re: We have added a new Figure 1, which depicts the molecular targets of the functionally investigated miRNAs in MM osteoblastogenesis and osteoclastogenesis processes. We wish to maintain the old Fig.1 (now Fig. 2), which graphically recapitulates the effects of ncRNAs, including exosomal ncRNAs, on the unbalance between osteoclastogenesis and osteoblastogenesis featuring MMBD.
6) Add the word 'multiple' in the title before 'myeloma'
Re: We thank the reviewer for this observation; we have added in the ‘title’ the word ‘multiple’ before ‘myeloma’.
Reviewer 2 Report
In the presented collaborative manuscript, authors have focused on the contribution of non-coding RNAs (ncRNAs) for the pathology of Multiple myeloma bone disease (MMDB). This comprehensive and well written review has been divided into several sections focusing on major signaling pathways and how these pathways are affected by loss and gain of ncRNA in MMDB. The text is well organized into paragraphs, allowing to find desired information yet each signaling pathway will be easier to follow if accompanied by graph visualizing them. Also, cytokines should be a wrapped as a table, same for ncRNAs.
In the current version of the manuscript there is only one figure and one table at the end of the article in the summary. The reader can learn from it that there is a disturbance between resorption and formation of bone mediated via ncRNA, but without any details. The table contain almost all ncRNAs and their role in MMDB are listed. Yet lncRNA TUG1 (potential biomarker of this disease) and MALAT1 (overexpression of which promote cell survival by regulating the expression and activity of the proteasome machinery) were not included in table. Inclusion and exclusion of ncRNA should be better rationalized.
In Notch signaling, four transmembrane receptors (Notch1-4) are described: yet only the function of Notch1, Notch2, Notch3 in osteoclastogenesis are described, it is not known what happens of is the role Notch4 – if there is no data this should be stated.
The paragraph about miR is very well written, divided into paragraphs, each describing the most important information about any given microRNA. The paragraph about lncRNA is written in a similar fashion yet it is not consistent, e.g. for some lncRNAs the position on the chromosome and length are listed, and for others not.
The article also has several typos, such as two spaces (e.g. line 69), or no space between the word and the reference (e.g. line 43).
Author Response
REVIEWER 2
In the presented collaborative manuscript, authors have focused on the contribution of non-coding RNAs (ncRNAs) for the pathology of Multiple myeloma bone disease (MMDB). This comprehensive and well written review has been divided into several sections focusing on major signaling pathways and how these pathways are affected by loss and gain of ncRNA in MMDB. The text is well organized into paragraphs, allowing to find desired information yet each signaling pathway will be easier to follow if accompanied by graph visualizing them.
Re: We wish to thank the reviewer for this recommendation. We have now added a new Figure 1, which depicts the molecular targets of the functionally investigated miRNAs regulating osteoblastogenesis and osteoclastogenesis in the myeloma bone marrow microenvironment, which are also summarized in Table 1 along with bone related effects and the tumor suppressive/promoting role of each of them in the context of MMBD.
Also, cytokines should be a wrapped as a table, same for ncRNAs.
Re: Thank you for this observation. We wish to remark to this referee that, in this review, we aimed to provide evidence of MMBD regulation by ncRNAs, and to accomplish this we have reported a new Figure (Fig.1) and a Table (Table 1) with the molecular targets of each ncRNA functionally involved in MMBD; therefore, we feel that a table on cytokines would divert the reader from the main focus of this review article.
In the current version of the manuscript there is only one figure and one table at the end of the article in the summary. The reader can learn from it that there is a disturbance between resorption and formation of bone mediated via ncRNA, but without any details. The table contain almost all ncRNAs and their role in MMDB are listed. Yet lncRNA TUG1 (potential biomarker of this disease) and MALAT1 (overexpression of which promote cell survival by regulating the expression and activity of the proteasome machinery) were not included in table. Inclusion and exclusion of ncRNA should be better rationalized.
Re: We thank the reviewer for this comment. As indicated in the title of the Table 1, we have reported in it only the ncRNAs functionally characterized in MMBD, for which MMBD-related target(s) has/have been identified.
Information on lncRNAs and circRNAs is mostly related to their prognostic role, and as we have stated in the relative sub-paragraphs, mechanistic studies are mostly missing and need to be performed. Therefore, we have included in the table only those ncRNAs for which a functional effect on MMBD-related molecules has been identified. We have clarified this relevant point in the paragraphs of lncRNAs, as well as in the conclusions section.
In Notch signaling, four transmembrane receptors (Notch1-4) are described: yet only the function of Notch1, Notch2, Notch3 in osteoclastogenesis are described, it is not known what happens of is the role Notch4 – if there is no data this should be stated.
We thank the reviewer for these comments; accordingly, we added a sentence in the manuscript (Page 2, line 88-93) pointing out that the notch pathway is deregulated in the MM; specifically, we underlined that only three receptors (Notch1, -2 and -3) and two ligands (Jagged 1 and -2) have been described in the disease. This result is not unusual, given that the deregulation of Notch signaling is described in several cancer types but frequently not all components of the signaling pathway are involved; notably, as reported in the manuscript, some of them can even play opposite effects depending on the cell context (PMID: 26308486; PMID: 30146333; PMID: 19524514)
The paragraph about miR is very well written, divided into paragraphs, each describing the most important information about any given microRNA. The paragraph about lncRNA is written in a similar fashion yet it is not consistent, e.g. for some lncRNAs the position on the chromosome and length are listed, and for others not.
Re: We thank the reviewer for this comment; we revised the paragraph about lncRNAs, trying to make it more complete and making the information more uniform, where possible.
The article also has several typos, such as two spaces (e.g. line 69), or no space between the word and the reference (e.g. line 43)
Re: We thank the reviewer for the useful comments; we revised the manuscript in order to correct typos.
Reviewer 3 Report
Raimondi et. al. reviewed non-coding RNAs in myeloma bone disease. They described the key signaling pathways and roles of different miRNAs and lncRNAs on multiple myeloma bone disease (MMBD) regulation. They conclude that these non-coding RNAs are promising molecules as novel therapeutic and prognostic tools in MMBD. Few minor comments:
- Table 1: It will be nice to add a column describing the non-coding RNA as oncogenic or tumor suppressor.
- It will be more informative if the authors describe other lncRNAs and cirRNAs that are involved in MMBD. A table describing lncRNAs and ciRNAs should be added.
Author Response
# REVIEWER 3
Raimondi et. al. reviewed non-coding RNAs in myeloma bone disease. They described the key signaling pathways and roles of different miRNAs and lncRNAs on multiple myeloma bone disease (MMBD) regulation. They conclude that these non-coding RNAs are promising molecules as novel therapeutic and prognostic tools in MMBD. Few minor comments:
Table 1: It will be nice to add a column describing the non-coding RNA as oncogenic or tumor suppressor.
Re: We thank the reviewer for this suggestion; we modified Table 1 reporting the oncogenic or tumor suppressor functions of ncRNAs with a described functional role in the context of MMBD.
It will be more informative if the authors describe other lncRNAs and circRNAs that are involved in MMBD. A table describing lncRNAs and ciRNAs should be added.
Re: Given the pleiotropic functions and the complex regulatory mechanisms underlying their activity, lncRNAs and circRNAs are more difficult to study and current information about their function is limited compared to miRNAs. In this review, we have reported available findings about all, although few, lncRNAs and circRNAs potentially involved in MMBD, which indeed deserve more in-depth functional characterization, as indicated in the conclusions.
As stated in the title of the Table 1, we have here reported only the functionally characterized ncRNAs, for which MMBD-related target(s) has(have) been identified. This table indeed includes two lncRNAs, namely HOXC-AS3 and MEG3.
It has to be noticed that available information on lncRNAs and circRNAs in MM is mostly related to their prognostic role or their effects in MM plasma cells, and-as we have remarked in the paragraphs relative to lncRNAs and circRNAs- mechanistic studies in the context of the MM microenvironment are mostly missing and need to be performed, along with appropriate lncRNA expression profiling studies, to provide a definitive scenario on lncRNAs activity in MMBD. This point has been appropriately remarked also in the conclusions.
Round 2
Reviewer 1 Report
The authors have answered all queries satisfactorily.